# Study on the Application of Modified Sn-Based Solder in Cable Intermediate Joints

**DOI:** 10.3390/ma15238385

**Published:** 2022-11-25

**Authors:** Wenbin Zhang, Ruikang Luo, Xuehua Wu, Chungang Xu, Chunguang Suo

**Affiliations:** 1Faculty of Mechanical and Electrical Engineering, Kunming University of Science and Technology, Kunming 650504, China; 2Faculty of Science, Kunming University of Science and Technology, Kunming 650504, China

**Keywords:** power cable, intermediate joint, Sn-based solder, In, Cu

## Abstract

With the increasing use of underground cables, the quantity and quality of intermediate joints demanded are also increasing. The quality of the traditional crimping intermediate joint is easily affected by the actual process of the operator, which may lead to the heating of the crimping part of the wire core, affecting the insulation performance of the cable, and finally causing the joint to break. However, aluminothermic reactive technology has some problems, such as a high welding temperature and an uncontrollable reaction. In order to solve these problems, according to the brazing principle and microalloying method, the optimal content of In in Sn-1.5Cu-based solder was explored, and then the connection of the middle joint of a 10 kV cable was completed using a connecting die and electrical connection process. The contact resistance and tensile strength of the joint were tested to verify the feasibility of this method. The results show that the maximum conductivity of the solder with 3.8% and 5% In content can reach 3.236 × 10^6^ S/m, and the highest wettability is 93.6%. Finally, the minimum contact resistance of the intermediate joint is 7.05 μΩ, which is 43% lower than that of the aluminothermic welded joint, and the tensile strength is close to that of the welded joint, with a maximum of 7174 N.

## 1. Introduction

With the gradual promotion of urbanization in various places, the proportion of underground cable laying is gradually increasing. Cross-linked polyethylene (XLPE) insulated cable has been widely used in urban power grids because of its convenience and stability [1]. XLPE cables usually have a service life of more than 50 years, but cable joints will lead to insulation aging failure [2,3] and insulation resistance reduction [4] under the action of thermal, electrical, mechanical, or microbial factors. With the continuous extension of distribution network lines, the use of intermediate joints is also increasing. According to the statistics, the line faults caused by cable joint accessories account for most of the total faults [5,6,7].

In order to reduce the line faults caused by cable joints, scholars from various countries have studied and analyzed the characteristics of cables, such as detecting the fault in a cable joint and locating it [8], or studying the characteristics of joint temperature, partial discharge, and insulation faults [9,10,11]. However, for the objectively existing faults in cable joints, monitoring can only reduce the occurrence of accidents and does not fundamentally solve the problem.

The current research shows that the causes of cable failure are more concentrated at the connection of intermediate joints. Due to the influence of construction technology on the quality of traditional crimping joints, such as low terminal fitting accuracy, uneven manual force, and unsmooth edges and corners of joints, etc., it is easy to cause a local temperature increase [12,13,14]. Solving the performance problem of intermediate joints can fundamentally reduce the failure of cable joints. Therefore, it is necessary to find a new joint connection technology.

Recently, Jongwuttanaruk et al. [15] reported on the use of response surface methodology to optimize the mechanical crimping process. Yu et al. [16] designed an electromagnetic pulse crimping device, which uses a strong pulsed magnetic field to act on the metal workpiece to produce plastic deformation to complete the joint connection. Li et al. [17] optimized the parameters of the electromagnetic pulse crimping midfield shaper. Dewi et al. [18] studied the exothermic welding technology, which basically solved the problem of electrical corrosion between dissimilar metals of intermediate joints, and the prepared joints had excellent electrical properties. However, it is more difficult to polish the joint on the spot.

Due to the health and environmental problems of traditional lead–tin solder, tin-based lead-free solder has been introduced. The new elements added to the tin-based system should meet some basic requirements [19,20,21], such as reducing the surface tension of the solder to improve its wettability, mechanical properties (e.g., mechanical fatigue, vibration, impact, and shear), and electrical properties (e.g., electrical conductivity and contact resistance). In order to solve the above problems, the soft brazing technology is based on Sn-Zn [22] and Sn-Cu [23] and adds materials such as Bi [24], In [25], and Ga [26] to improve the properties of the solder. However, most of the research on solder is more inclined to the field of electronic packaging and cannot be well applied to power cable connections. For example, when the eutectic point of Bi content in Sn-Bi [27] alloy is 58 wt.%, the melting temperature of solder is 139 °C, which cannot withstand the temperature rise under overload current. Although the melting temperature of eutectic Sn-Zn [28] solder is similar to that of tin–lead solder and has higher mechanical strength, zinc is easily oxidized, resulting in the formation of an oxide layer, and its wettability is not satisfactory, which brings higher requirements for subsequent welding. In the connection technology of medium- and high-voltage cables, the high wettability of solder is used to fill the gap in the cable strand to break the oxide film on the surface of the wire core and finally improve the mechanical and electrical properties of the joint.

Brazing technology can be applied in different application scenarios, and it is particularly important to produce a kind of brazing filler metal that meets the relevant conditions. In order to obtain a better ratio of solder, we first tested and analyzed the Sn-1.5Cu binary alloy, adding different amounts of In to modify it, and then used the mold and explored the joint connection process to complete the production of intermediate joints. The tensile strength and contact resistance of the joint were tested, and the mechanical and electrical properties of the aluminum hot fusion joint were compared and analyzed.

## 2. Materials and Methods

### 2.1. Method of Measuring Solder Properties

#### 2.1.1. Alloy Material Preparation and Ratio

According to the binary alloy phase diagram of Sn-Cu, the Sn-1.5Cu alloy material was prepared, and the thermal, electrical, mechanical, and interfacial properties were analyzed. By exploring the influence of In content on the properties of the substrate, excellent welding materials were selected. In the experiment, Sn-1.5Cu binary alloy was used as the basic material, and the pure Sn, Cu, and In particles of 99.99% metal elements were prepared by the laboratory. After calculating according to the percentage mass fraction, the corresponding metal elements were measured on an electronic scale and smelted in SY0002 melting furnace at 900 °C. KCL powder was added as a covering agent to isolate the air and reduce metal oxidation. During melting, the liquid metal needs to be fully stirred every other 15 min. After holding for 30 min, pour it into the mold and set it aside after solidification and cooling. This paper explores the effect of adding different amounts of In to Sn-1.5Cu-based solders, and the element composition of the alloy materials is shown in Table 1 below.

#### 2.1.2. Melting Point Test

The melting point of lead-free solder is one of the important standards to measure the material technology. In this experiment, the melting temperature was measured with a NETZSCH STA449F3 differential scanning calorimeter, and the alloy sample tested was about 20 g. Before the test, the sample needs to be cleaned with acetone and ultrasonic, and the surface oxide is removed. Secondly, the alcohol is cleaned and dried with 600 mesh sandpaper and finally put into the equipment for measurement. The whole experimental process needs to be filled with protective gas. The heating range of the experimental equipment is controlled at 20 °C to 250 °C, and the heating rate is set at 10 °C/min.

#### 2.1.3. Electrical Performance Test

Electrical conductivity is an important parameter that affects current transmission. The electrical conductivity testing device made by ULVAC Company in Japan was used to measure the properties of metal materials by four-point probe method, and the samples were prepared into cylindrical samples with a diameter of 6 mm and a height of 10 mm. The device directly tests the resistivity of the tested sample, so it needs to be calculated according to the formula. The conductivity is the reciprocal of the resistivity, and the expression of the conductivity is as follows: where *R* is the measured resistance, *S* is the area of the surface electrode, and t is the electrode thickness.
(1)$$\rho =\frac{RS}{t}\;$$

#### 2.1.4. Mechanical Property Test

Tensile samples are usually cut from the product or blank and machined, and linetypes, bars, and casting patterns with constant cross-sectional area can be tested without machining. Figure 1 shows the standard sample made by casting and its size (thickness: 2 mm). The mechanical properties of the standard samples were measured by the tension tester. Set the stretching rate at 2 mm/min.

#### 2.1.5. Wettability Test

The interface performance is also the wettability, which is mainly reflected in the combination of solid and liquid after contact. The wettability is determined by the wetting angle and spreading area. Spreading wetting: at a certain temperature, the spreading process of liquid solder on a solid surface depends on the ability of breaking the oxide film of flux and the capillary force and fluidity of liquid metal to realize the welding process of liquid metal and solid. The surface free energy of solders with different contents or metal compositions is different. The surface free energy determines the surface tension, and the change in surface tension determines the wettability and spreading performance.

The base material of the spreading wetting test was T3 copper, and its length, width, and height was 20 mm × 20 mm × 2 mm. L-2 flux of Guangdong Metal Research Institute of China was selected, JF-966C computer heating platform was used, and nitrogen was continuously put into the wetting test. The copper sheet was polished with sandpaper, cleaned with alcohol, and dried; 2.00 g ± 0.10 g solder was made into a spherical shape on the heating table, cooled, and ground to 1.80 g; under the protection of argon gas, the copper sheet was evenly smeared with flux, the solder ball was placed in the center of the copper sheet, and placed together on the heating table preheated at 280 °C; after waiting for the reaction, the surface was cleaned and photographed, image processing was carried out to calculate the spreading area, and the solder height was measured.

Figure 2 shows a schematic diagram of the wettability experiment. *S* is the spread area of the solder, and *H* is the height of the solder. The expression of the spreading rate of the solder is as follows, where *K*: spreading rate; H: solder height after spreading (mm); and S: solder spreading area (mm^2^):(2)$$K=\left(1-\frac{H}{1.24^{3}\sqrt{SH}}\right)\times 100\%$$

### 2.2. Manufacture and Testing Methods of Cable Intermediate Joint

#### 2.2.1. Manufacturing Method of Joint

This section takes the actual connection situation of 10 kV cable as the object and designs different sizes of molds to complete the middle joint connection of 120 mm^2^ (diameter of wire: 11.8 mm) cable. Figure 3 below shows a schematic diagram of the size of the cable intermediate joint. Figure 4 shows a schematic diagram of the brazing joint.

#### 2.2.2. Test Method for Intermediate Joints

The contact resistance is the main cause of the heating of the cable. when the contact resistance is reduced, the calorific value must be reduced. According to the electrical contact theory, when two contact elements come into contact, the seemingly smooth surface is actually uneven, and when the current passes through, it will always pass through the contact surface at the conductive spot, so the size of the contact surface determines the total resistance of the cable. At the same time, the electrical conductivity of the joint also reflects the ability of the solder to remove the oxide film of the conductor.

At present, the method of measuring contact resistance is commonly used to measure resistance voltage. The experimental test platform, as shown in Figure 5, polishes the copper wire core before testing to reduce the influence of surface oxide film on the resistance measurement. The cable tension test platform is shown in Figure 6. A large current generator is a special device to generate a large current, which has been widely used in various fields.

## 3. Research Results and Analysis

### 3.1. Overview

A joint size with excellent performance is obtained by testing and analyzing its electrical and mechanical properties. The influence of In content on the properties was analyzed, and the aluminothermic reactive welding joints were made and compared. From the test data, it is concluded that the brazing technology can be applied to the manufacture of cable intermediate joints, which is easier to control and safer than the brazing aluminothermic reaction.

### 3.2. Results of Performance Parameters of Solder

The melting point, tensile strength, electrical conductivity, and wettability of Sn-1.5Cu-XIn solder were tested according to the method described in the previous section, and the test results are shown in Table 2. With the increase in the In content, the melting point of solder decreases continuously, and because the melting point and melting range are smaller, the melting and solidification speed of the material is faster, which has a beneficial effect of shortening the construction period. The increase in In can inhibit the formation of malignant Cu_6_Sn_5_ intermediate compounds and improve the tensile strength of the solder. When the content of In is 5%, the material has the maximum tensile strength of 53.18 Mpa. In has a positive effect on the improvement of electrical conductivity, but when a small amount of In is added, it will destroy the metal bonds in the basic solder and cannot take full advantage of the high conductivity of copper. The addition of In can significantly improve the wettability of solder, and the maximum wettability can reach 93.6%. The spread of the solder is shown in Figure 7.

### 3.3. Effect of Temperature on Wettability of Solder

Because the electrical conductivity and strength values of Sn-1.5Cu-5In were slightly higher than those of 3.8% In solder, and the wettability was similar, the material was finally selected for the first stage of the process test. The wettability test of Sn-1.5Cu-5In was carried out at 230~280 °C and included three groups, as shown in Figure 8. The better the wettability, the better the ability to fill the strand gap in the joint in theory, so we explored the indirect effect of the welding temperature on the joint.

As shown in Figure 9, three groups of test results show that the wettability of Sn-1.5Cu-5In increases with the increase in temperature, which is about 18% higher than that at 230 °C. In order to explore the brazing process, a temperature of 280 °C, joint length L = 30 mm, diameter = 15mm, and gap c = 2 mm were selected in the experiment, and then the four factors were used as independent variables to complete the joint fabrication, and the cable joint data were measured.

### 3.4. The Influence of the Change in Joint Structure Parameters on Its Performance

In the process exploration, each joint was made three times, and the measurement result was the average. As shown in Figure 10, the best welding gap is 1 mm. When there is no gap on the end face of the joint, the fused solder cannot fully fill the gap of the cable strand, and the residue cannot be discharged in time after the flux reaction. With an increase in the joint gap, the rising trend of the contact resistance slows down.

As shown in Figure 11 and Figure 12, the performance of the joint was tested by changing the diameter and length of the cable joint. The results show that the diameter of the joint is basically linear to the increase in mechanical strength; the average 1 mm increases by 757.25 N, and the maximum tensile strength is 8368 N at D = 18 mm; and under the same trend, the joint has better electrical performance and lower resistance, which is also in line with the calculation law of conductor resistance. However, considering the difficulty of subsequent cable insulation recovery, when the larger joint diameter of the copper wire is excessive, the effective insulation thickness is smaller, which is not conducive to the insulation reliability of the joint. The diameter of the subsequent joint was made 16 mm. With the increase in joint length, the tensile strength is improved continuously, and the maximum tensile force is 7171 N in L = 30 mm. Because the electrical conductivity of copper is higher than that of solder, the increase in solder length has no obvious benefit for the electrical properties of the joint.

The electrical performance of welding at different temperatures is shown in Figure 13. The results show that, although the higher welding temperature will improve the wettability of the solder, the electrical properties of the fabricated joint will not change positively with the increase in temperature and have the lowest contact resistance at 250 °C. In order to analyze this phenomenon, the sections of joints at typical welding temperatures were sliced, as shown in Figure 14 for the sections of joints welded at 250 °C and 280 °C. It can be seen that the filling effect of solder in the strand is better at 250 °C, and there is basically no obvious gap or impurity residue. The reason is that the flux reacts more violently at high temperature, and the tiny bubbles cannot be discharged in time, which affects the actual flow area and finally leads to the change in resistance.

From the above results and analysis, the joint fabrication parameters are determined as follows: welding temperature: 250 °C, end clearance: 1 mm, joint diameter: 16 mm, and joint length: 30 mm.

### 3.5. Effect of In Content on Joint Properties

This section takes the actual connection of 10 kV cable as the object, according to the 120 mm^2^ specification cable, and designs the connection mold to complete the intermediate head connection in the following process order. (1) Strip off the insulating layer and semiconducting layer of the cable to expose the copper conductor; (2) polish the copper conductor and clean it with alcohol; (3) melt 300 g of solder at 270 °C, preheat the mold at 250 °C, and smear the flux on the surface of the wire core; (4) install the mold, adjust the end face of the wire core, and pour the liquid solder; and (5) cool, open the mold, and polish the joint. Figure 15a shows the brazed joint after cooling and opening the mold. Figure 15b shows an aluminothermic reactive fusion joint. 

After the welding process is determined in the previous section, the intermediate joint is made with Sn-1.5Cu-XIn (X = 0.8, 2, 2.8, 3.8, 5) solder. Symmetrically select the connector and the surrounding 30 mm, measure the positive and negative access to each test three times, and take the average experimental data as shown in Table 3. When the content of In is 5%, the minimum contact resistance is 7.05 μΩ, which has more advantages than the aluminum hot fusion joint in resistance. Because the diameter of the joint is slightly larger than the wire diameter of the conductor, the measured contact resistance is slightly lower than that of the copper conductor. As can be seen from Figure 7, the increase in the content of In in this range gradually reduces the contact resistance of the joint and improves the electrical properties of the joints, which is approximately linear. Moreover, the conductivity test of Section 2.1.3 on the solder shows that the electrical conductivity is strong when the content of In is 3.8% and 5%, which shows that the appropriate increase in In plays a positive role in the electrical properties of the joint.

The tensile strength of intermediate joints with different In content is shown in Table 4 below. When the In content is 5%, the tensile strength of the joint is the highest, which is 7174 N and almost 1600 N less than that of the aluminum hot-melt welded joint. In the actual operating environment, if there are enough reserved cable wires, the joint will not have a large load and will only need to meet the maximum pulling force of manual pulling, which is not greater than 3 kN.

Put the prepared joint sample into both ends of the fixture of the tension machine and set the drawing rate 2 mm/min. The tension curve of the aluminothermic reactive fusion joint and the brazed joint of 5% In is shown in Figure 16. The aluminum hot-melt welded joint can withstand greater tension; the elastic deformation stage basically reaches 4.9 mm, and the joint has no plastic deformation. For the tensile curve of 5% In joint, the elastic deformation stage is shorter, about 2.3 mm, and the plastic stage is longer.

## 4. Conclusions

In this paper, the application of Sn-1.5Cu-based solder in a cable intermediate joint was introduced in detail. We configured Sn-Cu-based solder and studied different amounts of In to modify it to obtain a better performing solder composition. Then, taking 10 kV (120 mm^2^) cable as the object, a better welding process was developed by adjusting the temperature, gap, diameter, and length. Based on this process, the joint fabrication of Sn-1.5Cu-XIn solder was completed. After testing the contact resistance and tensile strength of the joint, we arrive at the following main conclusions:(1)Solder properties: The alloy solder is based on the Sn-1.5Cu system. By adding In to modify it, we found that with the increase in In the melting point decreases from 227.6 °C to 220.7 °C. The tensile strength of solder increases and can reach 57.18 Mpa when the content of In is 5%. The electrical conductivity of the solder increases slowly when the In content is 3.8%, and the solder conductivity reaches the maximum value of 3.236 × 10^6^ S/m. When the In content is 3.8%, the wettability spreading area is the largest. The value is 93.05 mm^2^, and the wettability is 93.6%.(2)Joint manufacturing technology: In terms of joint size, we have studied a diameter range of 14 mm~18 mm, length range of 10 mm to 30 mm, welding temperature range of 230~280 °C, and welding gap of 0~4 mm. From the test results, the process is determined as follows: diameter: 16 mm, length: 30 mm, temperature: 250 °C, and welding gap: 1 mm.(3)Joint performance: The results show that the contact resistance of the joint decreases with the increase in In content, the contact resistance changes approximately linearly, and the minimum contact resistance is 7.05 μΩ when the In content is 5%. Compared with the aluminum hot-welded joint (12.39 μΩ), it has better electrical properties; the increase in In content in this range can improve the tensile strength of the joint, and the maximum tensile strength is 7174 N at 5%, which is close to 8797 N.

At present, there are still great difficulties in manufacturing power cable joints in harsh environments, and the improper manufacturing of copper wire joints will bury serious hidden dangers in the transmission lines. In this paper, by comparing and testing the performance of the aluminothermic reactive fusion joint in the more mature market, the developed brazing method was shown to achieve an excellent connection effect in the cable core connection. In the follow-up, it is necessary to analyze and demonstrate the reliability of the long-term operation of the wire core joint.

## Figures and Tables

**Figure 1 materials-15-08385-f001:**
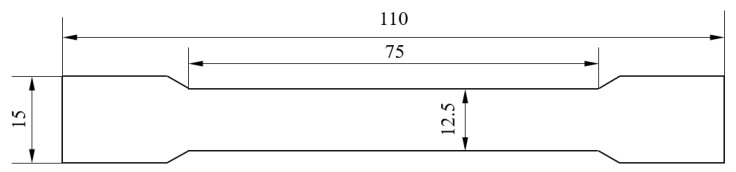
External dimensions of standard samples produced by pouring. (unit: mm).

**Figure 2 materials-15-08385-f002:**
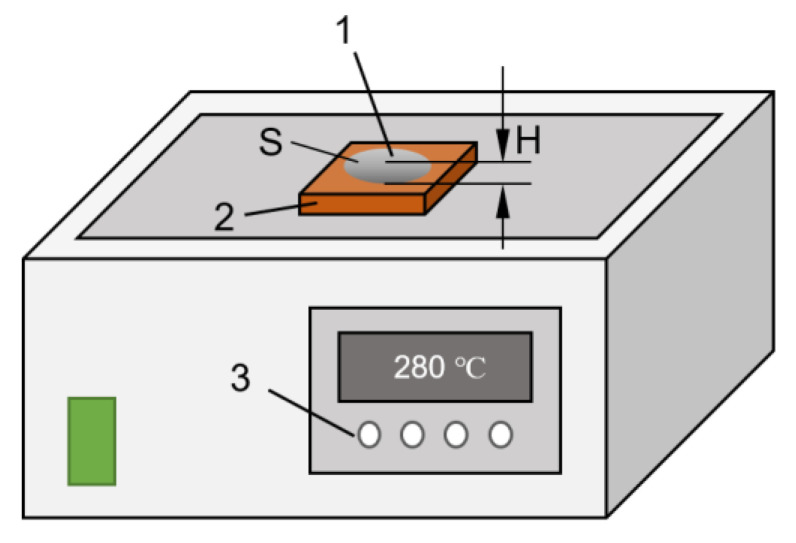
Schematic diagram of wettability test platform: 1—solder material; 2—copper sheet; 3—temperature controller.

**Figure 3 materials-15-08385-f003:**
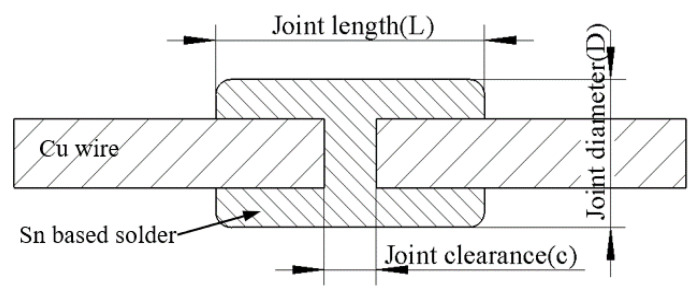
Dimensions of cable intermediate joint.

**Figure 4 materials-15-08385-f004:**
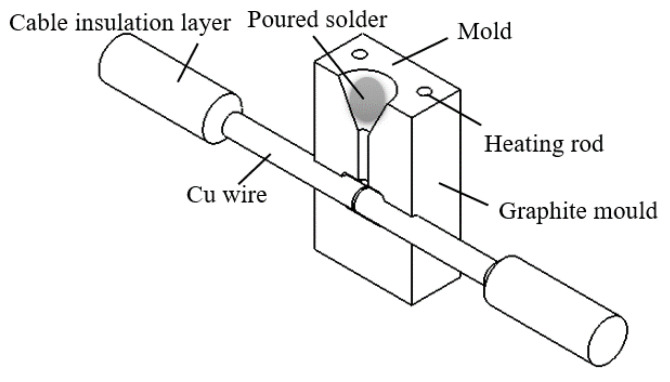
Diagram of the brazing joint.

**Figure 5 materials-15-08385-f005:**
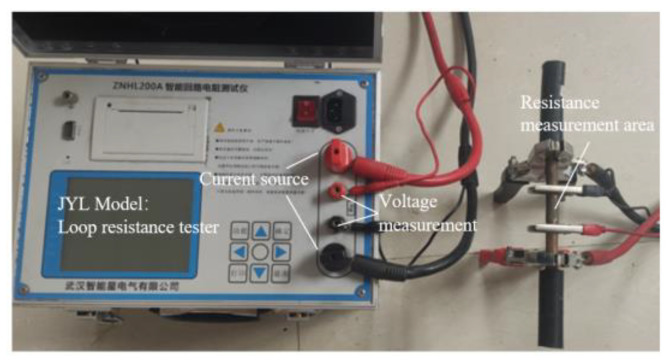
Contact resistance test platform.

**Figure 6 materials-15-08385-f006:**
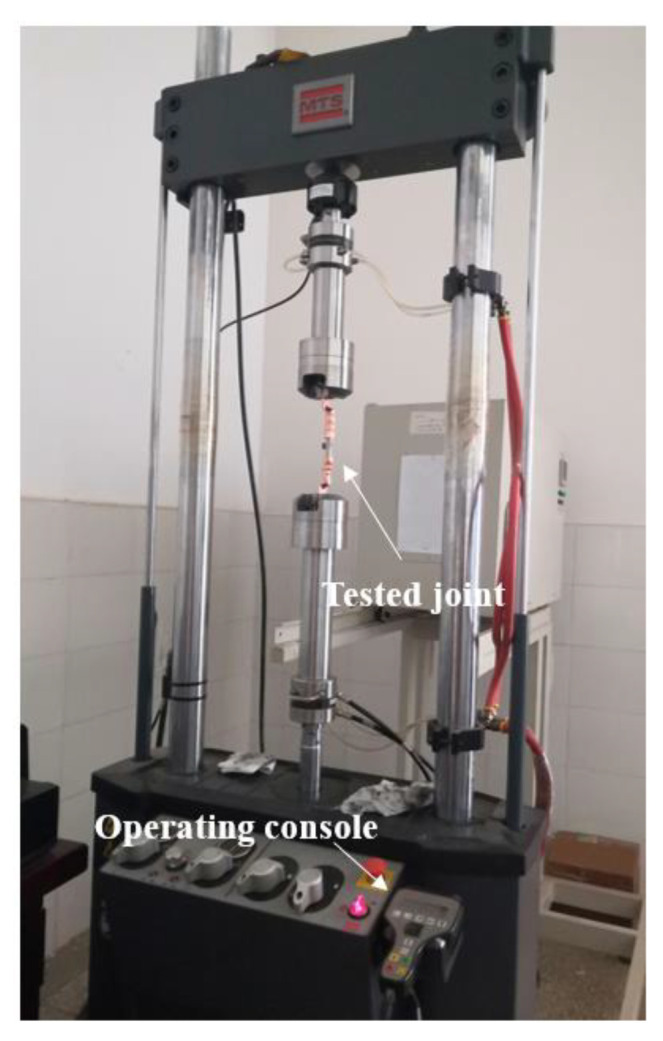
Tensile test platform for joint.

**Figure 7 materials-15-08385-f007:**
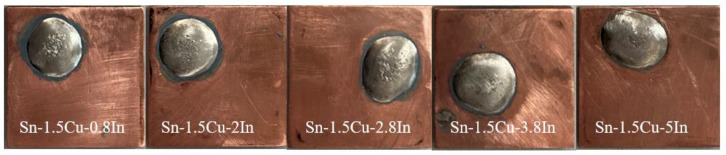
Spreading wetting of solder on copper sheet.

**Figure 8 materials-15-08385-f008:**
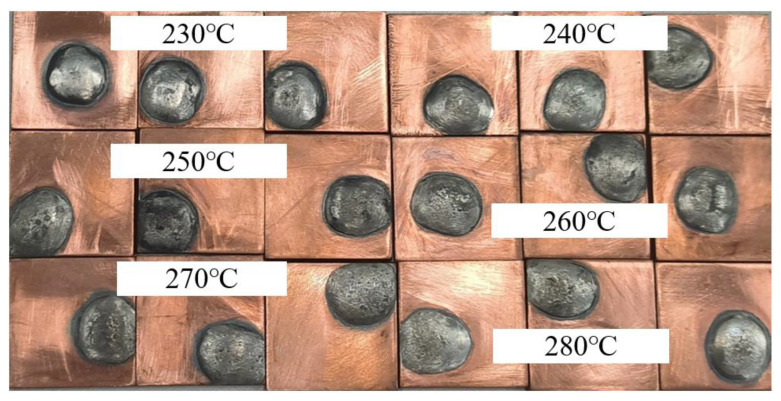
Spreading of Sn-1.5Cu-5In solder at different temperatures.

**Figure 9 materials-15-08385-f009:**
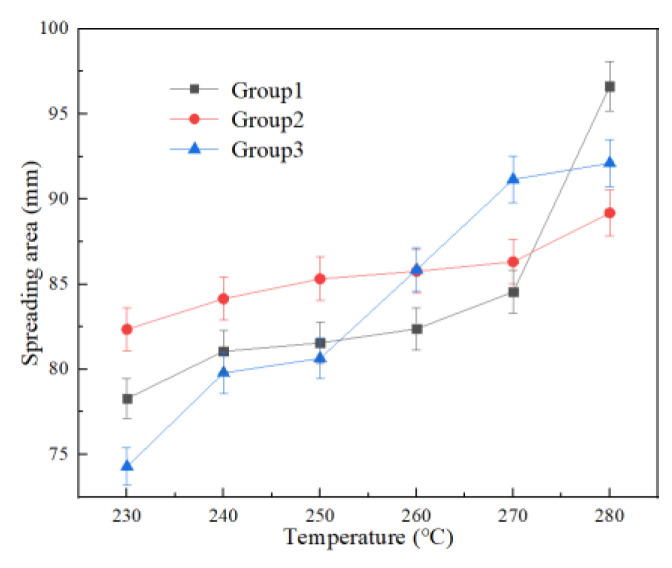
Effect of temperature on wettability of Sn-1.5Cu-5In solder.

**Figure 10 materials-15-08385-f010:**
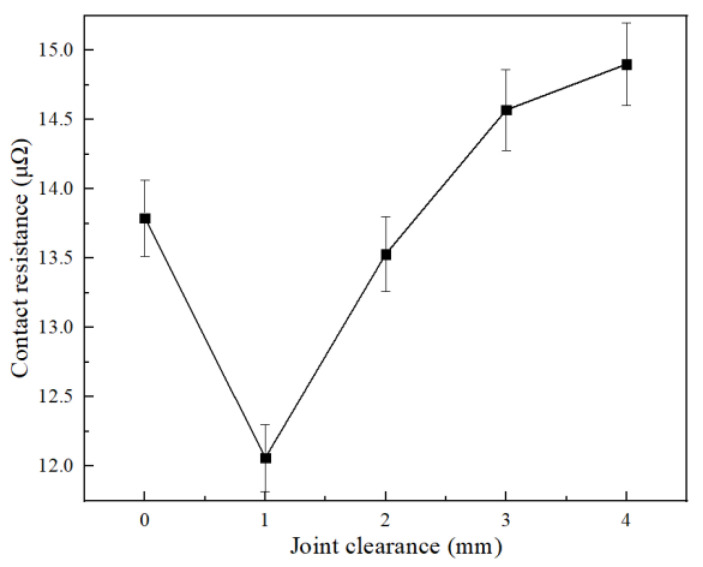
Effect of welding gap on joint resistance.

**Figure 11 materials-15-08385-f011:**
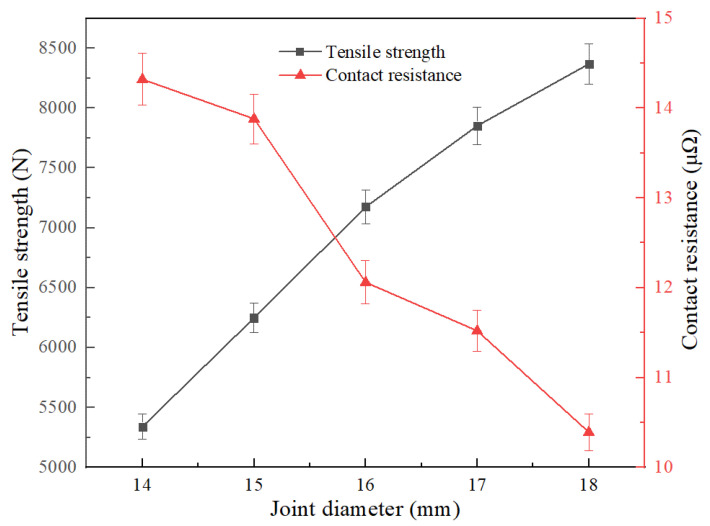
Effect of welding diameter on resistance and strength of joint.

**Figure 12 materials-15-08385-f012:**
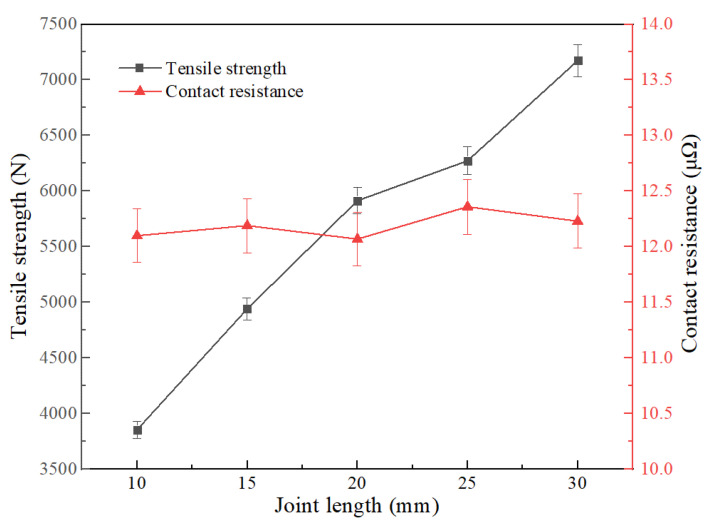
Effect of welding length on resistance and strength of joint.

**Figure 13 materials-15-08385-f013:**
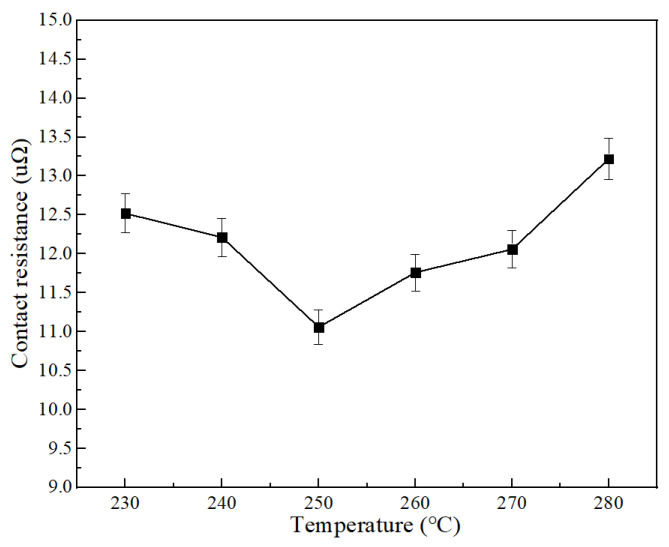
Effect of welding temperature on joint resistance.

**Figure 14 materials-15-08385-f014:**
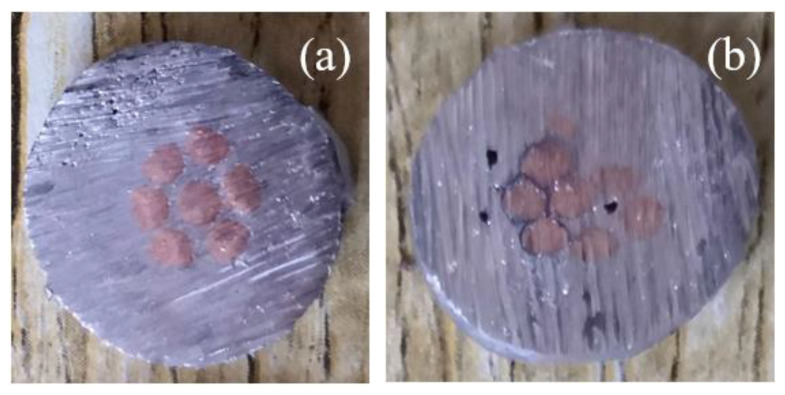
Cross section of cable joints: (**a**) 250 °C, (**b**) 280 °C.

**Figure 15 materials-15-08385-f015:**
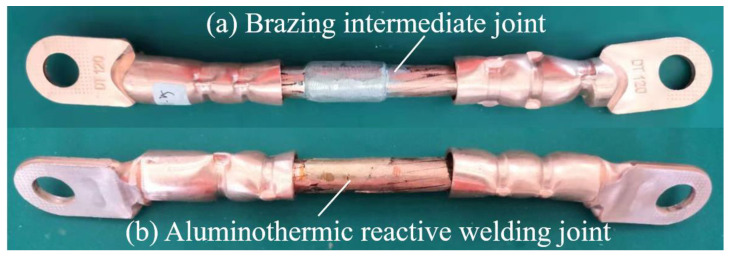
Physical object of polished intermediate joint.

**Figure 16 materials-15-08385-f016:**
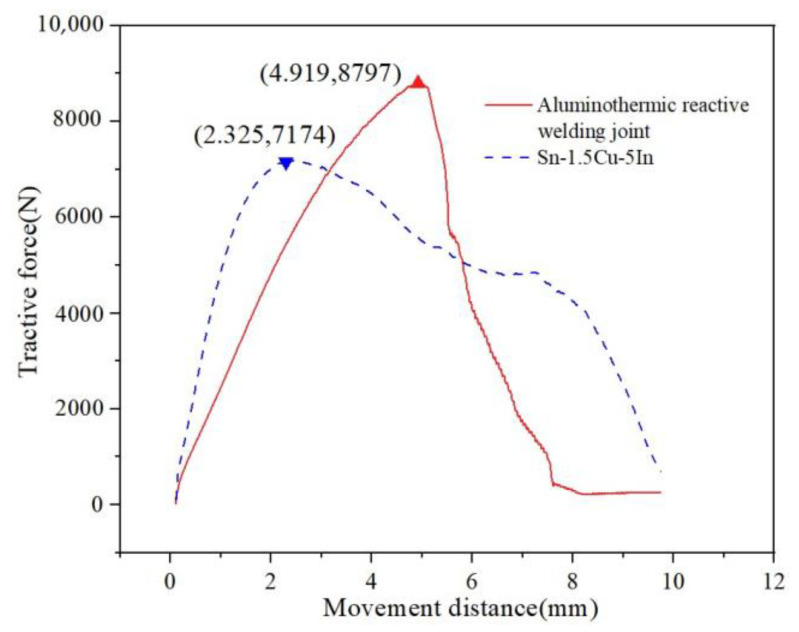
Mechanical properties of intermediate joints (red wire: thermite joint; blue line: Sn-1.5Cu-5In).

**Table 1 materials-15-08385-t001:** Alloy composition of solder.

Alloy Composition	Sn (g)	In (g)	Cu (g)
Sn-1.5Cu-0.8In	97.7	0.8	1.5
Sn-1.5Cu-2In	96.5	2	1.5
Sn-1.5Cu-2.8In	95.7	2.8	1.5
Sn-1.5Cu-3.8In	94.7	3.8	1.5
Sn-1.5Cu-5In	93.5	5	1.5

**Table 2 materials-15-08385-t002:** Performance parameters of solder.

Alloy Composition	Melting Point (°C)	Melting Range (°C)	Tensile Strength (MPa)	Electric Conductivity (S/m)	Spreading Area (mm^2^)	Wetting Rate (%)
Sn-1.5Cu	232.4	4.7	20.54	2.841 × 10^6^	49.88	87.6
Sn-1.5Cu-0.8In	227.6	4.1	37.50	2.224 × 10^6^	58.19	89.4
Sn-1.5Cu-2In	225.5	3.5	41.92	2.387 × 10^6^	67.53	90.7
Sn-1.5Cu-2.8In	224.3	2.9	43.86	2.532 × 10^6^	85.65	92.1
Sn-1.5Cu-3.8In	223.0	2.4	52.31	3.195 × 10^6^	93.05	93.6
Sn-1.5Cu-5In	220.7	2.0	53.18	3.236 × 10^6^	92.64	93.3

**Table 3 materials-15-08385-t003:** Test results of contact resistance of intermediate joints.

Joint Category	Contact Resistance (μΩ)
Copper wire	7.88
Sn-1.5Cu	15.90
Sn-1.5Cu-0.8In	14.37
Sn-1.5Cu-2In	12.16
Sn-1.5Cu-2.8In	10.73
Sn-1.5Cu-3.8In	7.59
Sn-1.5Cu-5In	7.05
Aluminothermic reactive welding joint	12.39

**Table 4 materials-15-08385-t004:** Test results of tensile strength of intermediate joints.

Joint Category	Maximum Pulling Force (N)
Sn-1.5Cu-0.8In	6082
Sn-1.5Cu-2In	6217
Sn-1.5Cu-2.8In	6435
Sn-1.5Cu-3.8In	6810
Sn-1.5Cu-5In	7174
Aluminothermic reactive welding joint	8797

## Data Availability

The data used to support the findings of the study are available from the corresponding author upon request (kangruiluo@163.com).

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
