# Peer review of "Study on the Application of Modified Sn-Based Solder in Cable Intermediate Joints"

_materials, 2022, doi:10.3390/ma15238385_

Round 1
Reviewer 1 Report
The paper under review is devoted to a hot topic, namely the use of lead-free solders for connecting cables laid in urban areas in underground utilities. The article will be useful to readers, but now it has flaws. I hope my recommendations will help the authors to publish an article in this scientific journal.
1. In the introduction, you need to give more information about the need to use lead-free solders, indicate what caused this. The article https://doi.org/10.3390/machines9050093 has information on this.
2. There is a mistake in the word "Figure". The letter "r" is missing and the wrong type of font is selected.
3. In Figure 3 and 8, I recommend that you indicate the size of the solder in each photo. You can also sign the content In on the photo. Now these photos for the reader have no information and meaning.
4. You need to change the names of all Figures. Now they are very uninformative and do not reflect the essence of the information presented in a graph, diagram or photo.
5. The material of the paper is very useful, but now it looks unfinished. It will be better and more familiar for the reader if the authors present separately information about the materials and methods of research in section 2. Materials and methods, and Research results and their analysis in section 3. Results and discussion. Now everything is mixed up and information is difficult to perceive.
Author Response
Please see the attachment,I marked out my changes to your comments in the word review.
Attention review 1 please

Reviewer 2 Report
2.2.3. Mechanical property test
- Please insert the thickness of the tensile sample.
2.3.1. Test material ratio
- If it is possible, please insert the chemical compositions after solidification.(ex. ICP results)
Fig 9, 10, 11, 12, 13
- Please insert the error bars.
Author Response
Please see the attachment,I marked out my changes to your comments in the word review.
Attention review 2 please

Reviewer 3 Report
The authors studied the application of modified Sn-based solder in cable intermediate joint. This is a good study and I will like to say well done to the research team. However, the below listed minor comments should be addressed:
Abstract
- Please rewrite the capitalized word in this statement: “…aluminothermic WELDING joint…” as “…aluminothermic WELDED joint…”
Introduction
- Kindly remove initials of authors from the text citation. For instance, see “K. Jongwuttanaruk et al. [15]”
Section 2
- Specify the material of the mold
- Specify the type of protective gas used for the experiment
Section 3
- Specify the size of brazed Cu wire
Author Response
Please see the attachment,I marked out my changes to your comments in the word review.
Attention review 3 please

Round 2
Reviewer 1 Report
I recommend the new version of the article for publication in the scientific journal Materials.
Author Response
Response to Reviewer 1 Comments
I have adjusted the formula on your version and aligned it. I have made minor modifications to the text and picture layout to meet the reading needs.Such as Review1(secondly)
Thanks
